# Anti-CRISPR *Anopheles* mosquitoes inhibit gene drive spread under challenging behavioural conditions in large cages

Rocco D'Amato[1], Chrysanthi Taxiarchi [2], Marco Galardini [3,4,5], Alessandro Trusso[1], Roxana L. Minuz[1], Silvia Grilli[2], Alastair G. T. Somerville[2], Dammy Shittu[2], Ahmad S. Khalil [3,6,7], Roberto Galizi [8], Andrea Crisanti [2,9], Alekos Simoni [1,2,11] ✉ & Ruth Müller[1,10,11] ✉

CRISPR-based gene drives have the potential to spread within populations and are considered as promising vector control tools. A *doublesex*-targeting gene drive was able to suppress laboratory *Anopheles* mosquito populations in small and large cages, and it is considered for field application. Challenges related to the field-use of gene drives and the evolving regulatory framework suggest that systems able to modulate or revert the action of gene drives, could be part of post-release risk-mitigation plans. In this study, we challenge an AcrIIA4-based anti-drive to inhibit gene drive spread in age-structured *Anopheles gambiae* population under complex feeding and behavioural conditions. A stochastic model predicts the experimentally-observed genotype dynamics in age-structured populations in medium-sized cages and highlights the necessity of large-sized cage trials. These experiments and experimental-modelling framework demonstrate the effectiveness of the anti-drive in different scenarios, providing further corroboration for its use in controlling the spread of gene drive in *Anopheles*.

The need for sustainable methods addressing the burden of mosquito-borne pathogens advanced the development of CRISPR/Cas9-based gene drive mechanisms following two different approaches: population suppression, where the gene drive reduces the number of mosquitoes in a population, or population replacement, in which the drive aims to imprint a favourable trait into the population[1–3]. Over a decade there has been a great progress in the development of CRISPR-based self-sustaining gene drives as a promising potential method for effectively controlling mosquito vectors[2–7]. To date, these systems have been tested only in laboratory settings, in combination with mathematical modelling studies, demonstrating the efficacy and feasibility for population control of malaria-transmitting mosquito species[7,8]. WHO guidelines for testing GM mosquitoes[9] advise for a step-wise approach of technology development, where the mosquito strains are tested in sequential settings of incremental complexity. For instance, an initial evaluation of efficacy is performed in single generation experiments to evaluate the bias of inheritance and the main fitness parameters (fecundity, longevity, mating ability, etc). This is usually

[1]Genetics and Ecology Research Centre, Polo of Genomics, Genetics and Biology (Polo GGB), Terni, Italy. [2]Department of Life Sciences, Imperial College London, London, UK. [3]Biological Design Center, Boston University, Boston, MA, USA. [4]Institute for Molecular Bacteriology, TWINCORE Centre for Experimental and Clinical Infection Research, a joint venture between the Hannover Medical School (MHH) and the Helmholtz Centre for Infection Research (HZI), Hannover, Germany. [5]Cluster of Excellence RESIST (EXC 2155), Hannover Medical School (MHH), Hannover, Germany. [6]Department of Biomedical Engineering, Boston University, Boston, MA, USA. [7]Wyss Institute for Biologically Inspired Engineering, Harvard University, Boston, MA, USA. [8]Centre for Applied Entomology and Parasitology, School of Life Sciences, Keele University, Keele, UK. [9]Department of Molecular Medicine, University of Padova, Padua, Italy. [10]Unit of Entomology, Department of Biomedical Sciences, Institute of Tropical Medicine, Antwerp, Belgium. [11]These authors contributed equally: Alekos Simoni, Ruth Müller. ✉e-mail: a.simoni@imperial.ac.uk; r.muller@itg.be

followed by cage population testing, where a gene drive is introduced into a target population in small cages and its ability to increase in frequency is monitored in consecutive, discrete generations. The subsequent step may involve testing under physical confinement within a large cage that simulates the disease-endemic setting, as was the case for the *Anopheles gambiae* Ag(QFS)1 gene drive strain[5].

Potential applications of gene drives for vector control bring also regulatory challenges for testing of gene drive mosquitoes in the field[10–13]. Highly effective gene drives could spread over large areas in a relatively short timeframe or spread beyond the targeted areas. Such events could potentially be mitigated by modulating or inhibiting the activity of CRISPR-Cas9 to prevent the gene drives from spreading or to revert their activity in certain contexts. Other than inhibition of gene drive as a result of genetic resistance at the target site[3,14], several engineered systems have been proposed to counteract the action of CRISPR-based gene drives, following two main approaches: DNA cleavage of the drive allele[15–18], or by cleavage-independent protein inhibition of the Cas9 nucleases[19–22]. The latter is achieved through the use of the naturally occurring anti-CRISPR (Acr) proteins, products of the evolutionary arms race between bacterial adaptive immune systems and bacteriophages[23]. Acrs have been used in several heterologous eukaryotic systems for temporal and spatial post-translational control of CRISPR-mediated gene editing, offering great potential for improvement of therapeutic uses of CRISPR[24,25]. Other than mammalian cells in which the functionality and adaptability of Acrs has been extensively researched, Acrs have also been proved effective in CRISPR-Cas inhibition in higher organisms, such as plants[26], mice[27] and zebrafish[28].

Anti-drive strategies based on DNA cleavage were shown effective in stopping Cas9 activity and even replacing gene drive from lab populations[18–20]. An Acr-based system for gene drive inhibition can have additional benefits compared to approaches based on DNA cleavage. First, it could act against any Cas9-based gene drive, and it does not need to be tailored to specific strains. Although this could impact several gene drive interventions, and it cannot be restricted to individualised needs, it offers great potential as a risk mitigation strategy that could be widely used. Secondly, unlike cleavage-based countermeasures, Cas9-inhibition via protein interaction does not generate various recombinant events, therefore its outcome is more predictable and off-target effects are minimal.

Previously, a transgenic mosquito named Ag(Vasa:A4) was generated expressing the anti-CRISPR peptide AcrIIA4 under a germline-specific promoter in *An. gambiae*[21]. The Ag(Vasa:A4) construct is inherited following Mendelian rules and its persistence in a population therefore is dependent on its fitness costs. It was demonstrated that a single release of the Ag(Vasa:A4) strain, that for simplicity we called here 'anti-drive', was able to block the highly active *doublesex* gene drive Ag(QFS)1[7], showing 100% inhibition of homing and the ability to prevent population collapse in a small-caged mosquito population[21]. Due to fitness cost, likely associated to the integration site of the genetic construct, and the mode of protein-based action not genetically excising (directly of by recombination) the gene drive construct, the anti-drive Ag(Vasa:A4) strain was unable to remove the gene drive from the population.

In advance of any considerations of anti-drive strategies for field applications, it is required to have an in-depth characterisation of efficacy and impact in lab contained populations. Complex large population studies with overlapping generations can provide valuable information when assessing and evaluating the efficacy or possible limitations of genetic control strategies. Therefore, the aim of the present study is to test the functionality and the predictability of an anti-drive system in *An. gambiae,* when transgenic mosquitoes are exposed to intricate behavioural conditions, in contained lab settings. We used the previously developed AcrIIA4-based system to generate a second anti-drive strain, that consistently inhibited Cas9-induced

cleavage and homing and showed lower fitness costs than the previous one. We created an agent-based stochastic model to predict genotype dynamics (anti-drive, gene drive, wild-type) in age-structured mosquito populations and the impact of various anti-drive fitness costs in a range of release size scenarios. We tested model predictions by releasing the anti-drive strain into medium and large cages comprising a mix of wild-type and gene drive mosquitoes. The anti-drive strain was able to prevent the spread and to induce the decline and even elimination, in one case, of the suppressive Ag(QFS)1 gene drive in age-structured overlapping populations that involve more complex behaviour and ecological conditions. Our findings provide evidence that the anti-drive system is highly effective and highlight the value of large cage testing for the evaluation of mosquito strains prior to field testing.

## Results
### Generation and selection of a new anti-drive strain
A CRISPR-based gene drive relies on the expression of the Cas9/gRNA complex in the germline to promote site-specific cleavage and homologous-dependent DNA repair in a process called 'homing', which ensures a super-Mendelian inheritance of the construct (Fig. 1A, left panel). We previously generated an *An. gambiae* transgenic strain, named Ag(Vasa:A4), that express the anti-CRISPR AcrIIA4 protein in the germline under the control of the *vasa2* promoter[21,29]. In trans-heterozygous Ag(QFS)1[+/−]/Ag(Vasa:A4)[+/−] progeny the AcrIIA4 peptide was shown to inhibit Cas9 activity, resulting in inheritance of both constructs in a Mendelian fashion (Fig. 1A, right panel)[21]. However, the Ag(Vasa:A4) line showed reduced fitness compared to wild-type, likely associated with the integration locus of the genetic construct[21]. Moreover, the fitness costs were exacerbated when homozygote males and females were crossed to each other, the egg output was drastically reduced, and it was not possible to maintain the strain in homozygosity (Fig. 1B).

Since the population dynamics of gene drive and anti-drive mosquitoes depends on their own efficacy (homing rate and homing inhibition, respectively) and relative fitness[30], we aimed to generate an anti-drive transgenic line with minimal fitness costs. To do so, we designed a genetic construct carrying the vasa2::AcrIIA4 and 3xP3::eGFP expression cassettes flanked by *piggyBac* arms (Supplementary Fig. 1A), to allow its random transposon-mediated insertion into the mosquito genome. The transgenic founders (named Ag(Vasa:A4)2), obtained from individuals injected with that construct, were crossed to the Ag(QFS)1 mosquitoes to select lines of interest based on gene drive inhibition and fitness. Three lines had the construct inserted in same genomic location, while a fourth one was harboured in a different genomic site (Supplementary Table 1); all of them were characterised by a complete homing inhibition and high larval output (Supplementary Fig. 1B). However, mosquitoes from only one line (with a transgene located on the 2 R chromosome, within the first intron of the AGAP004649 gene, and ~10 megabases from the *dsx* locus) were able to survive to adulthood when in homozygosity, and that line was chosen for further phenotypic characterization.

### Ag(Vasa:A4)2 anti-drive line showed improved life-history parameters compared to previous strain
The inheritance rate of the anti-drive allele from the Ag(Vasa:A4)2[+/−] individuals was in agreement with Mendelian rates of inheritance (average of 48% GFP positive from males and 52% from females), as expected (Fig. 1C). No significant fertility or fecundity cost associated with the presence of the anti-drive construct was observed in either hetero- or homozygote male or female mosquitoes when compared to the wild type (Supplementary Fig. 2A, $p > 0.99$; Kruskal–Wallis's test). We also measured the time of pupation, and larval and pupal survival rates for each sex and genotype (see Methods, Supplementary Tables 2 and 3). Similarly, the survival of the aquatic stages (larvae and pupae)

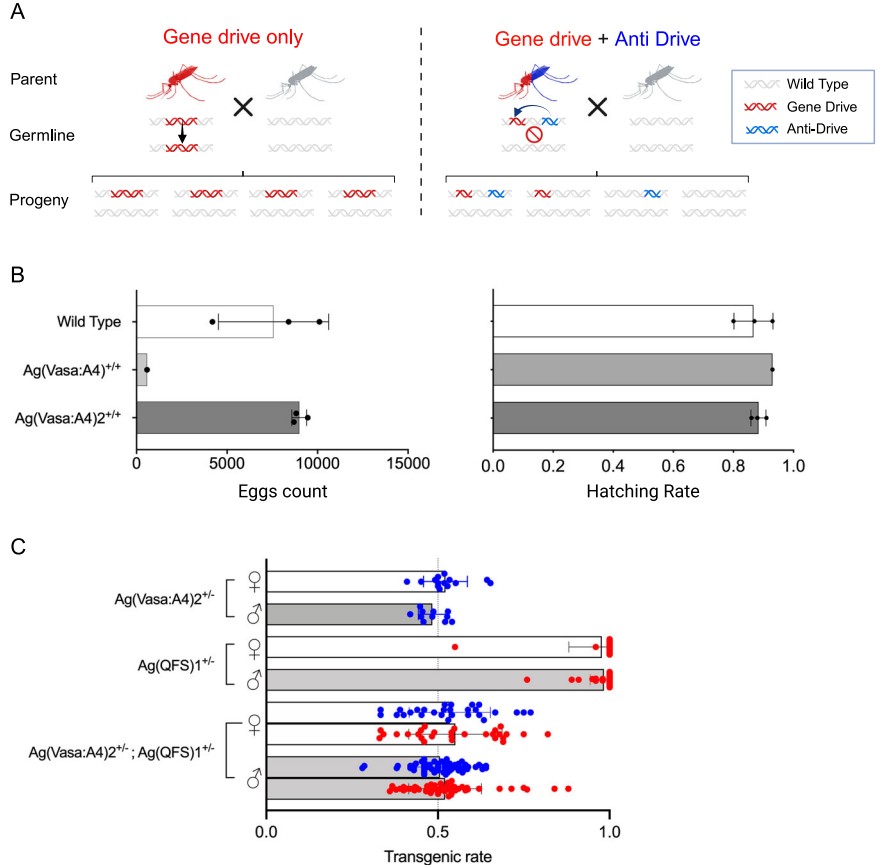

**Fig. 1 | Improved fitness and consistent blocking of gene drive homing in the updated anti-drive strain. A** Heterozygous Ag(QFS)1 gene drive (in red) crossed to WT individuals (in grey) produce progeny most of which inherits the drive allele through homing (indicated with the black arrow) in the germline (left-hand panel). Trans-heterozygous individuals carrying one allele of the Ag(QFS)1 drive and one of the Ag(Vasa:A4)2 anti-drive (in red and blue, respectively) crossed to WT result in Mendelian inheritance of both the gene drive and anti-drive alleles (right-hand panel) due to blocking of Cas9 in the germline. **B** Dot plots overlapping bar plots of the number of eggs and the relative hatching rate calculated from bulk oviposition of homozygous males and females Ag(Vasa:A4)[+/+] (light grey bar) and Ag(Vasa:A4) 2[+/+] (dark grey) compared to wild-type control (white). There were no statistically significant differences between hatching rates, while there was for the egg number between the Ag(Vasa:A4)[+/+] and the wild type (*p* = 0.458, Kruskal−Wallis test). Bar charts indicate mean percentage values and error bars indicate standard error of the mean from all biological samples assessed for each cross. Three biologically independent experiments were examined for each cross, consisting of 100

ovipositing females in each cross. The respective raw data are provided in the Source Data file. **C** Scatter plot showing the transgenic rate from individual deposition of Ag(QFS)1 (red dots) and the Ag(Vasa:A4)2 (blue dots) alleles in the progeny from the crosses of male and female anti-drive heterozygotes, gene drive heterozygotes, and drive/anti-drive trans-heterozygous for the two alleles, with the corresponding wild-type. Both male and female trans-heterozygotes showed complete inhibition of homing and Mendelian rates of the drive allele inheritance (*p* < 0.0001, Kruskal−Wallis test). Bar charts (light grey for males and white for females transgenics) indicate mean percentage values and error bars indicate standard error of the mean of transmission rates from all biological samples assessed for each cross. A minimum of 11 biologically independent samples (ovipositing females) were examined per independent experiment. Two independent experiments were performed for drive/anti-drive trans-heterozygotes, one experiment was performed for the anti-drive heterozygotes, and the data for gene drive heterozygotes were sourced from[7]. The respective raw data are provided in the Source Data file.

and the time of pupation was comparable among the strains (Supplementary Table 3, *p* > 0.99; Kruskal−Wallis's test; Supplementary Table 2, *p* = 0.88, Mann-Whitney test). We observed a small reduction in mating capacity of the homozygote anti-drive males when compared to wild-type males (Supplementary Table 4, *p* = 0.0407; Kruskal−Wallis test), suggestive of a moderate fitness cost associated with carrying two copies of the anti-drive construct by male mosquitoes. We observed a higher 50% median adult longevity in medium-sized cages for both heterozygote and homozygote anti-drive males (19 and 16 days respectively) and females (26 days) compared to wild-type individuals (14 days for males and 21 days for females), but same overall longevity (Supplementary Fig. 3). However, adult longevity in the large-sized cages was markedly reduced (50% median mortality of 5−6 days), compared to adults from the medium-sized cages. No statistical difference in median adult survival was observed in large cages for heterozygous or homozygous males or females compared to control (Supplementary Fig. 4, log-rank Mantel-Cox).

## Synergistic effect of drive and anti-drive transgenes on fitness parameters

To test the inhibitory activity of homing and the fitness impact of the Ag(Vasa:A4) transgene on gene drive performance, we crossed the Ag(QFS)1[+/−];Ag(Vasa:A4)2[+/−] trans-heterozygotes to wild-type and assessed fertility and transmission frequencies of drive (D) and anti-drive (A) alleles in single deposition assays (Fig. 1C). Male and female individuals heterozygous for the gene drive construct were also crossed to wild-type mosquitoes and assessed in parallel. Individuals carrying both drive and anti-drive transgenes showed an average 52% and 55% rate of the drive inheritance from males and females respectively, indicating complete inhibition of homing, in contrast to 96% and 99% rate of inheritance of the gene drive alone (Fig. 1C and Supplementary Tables 8 and 9). While the Ag(QFS)1 strain exhibits reduced female fertility in heterozygosity[7] (Supplementary Fig. 2), the presence of the anti-drive construct increased the fecundity of the trans-heterozygous individuals expressing the gene drive, as reported previously[21]. The

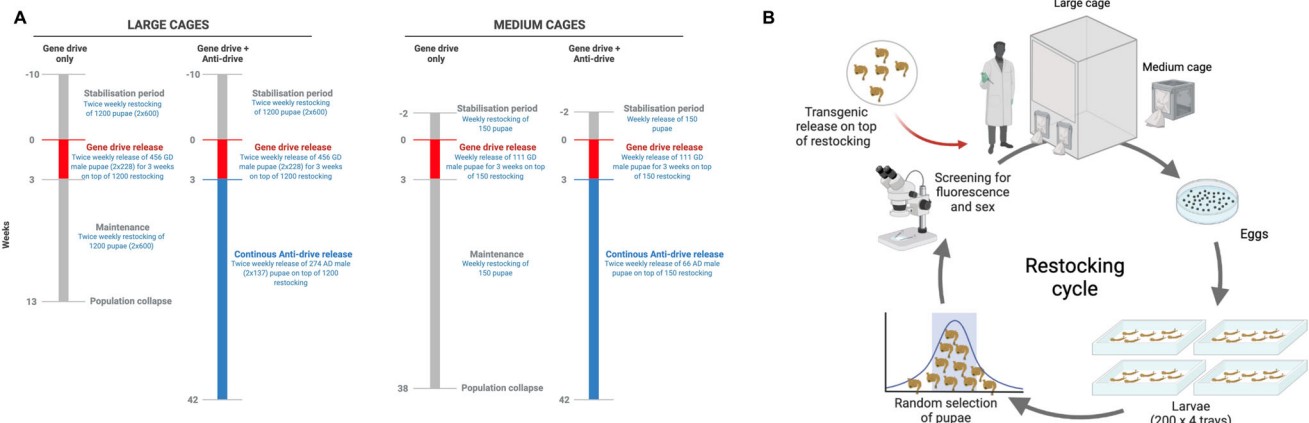

**Fig. 2 | Timeline of the anti-drive release experiment in medium and large-sized cages. A** Schematic outline of the large and medium-sized cage experiments, indicating the different experimental steps, in weeks, and the time of releases of gene drive (in red) and anti-drive (in blue). Number of weekly restocked and released mosquitoes is indicated. **B** Schematic representation of the pupae restocking cycle which occurred twice a week in the large cages, and once a week in the medium-sized cages. During the maintenance phase, no transgenic mosquitoes were added to the cage and the population was self-maintained by restocking.

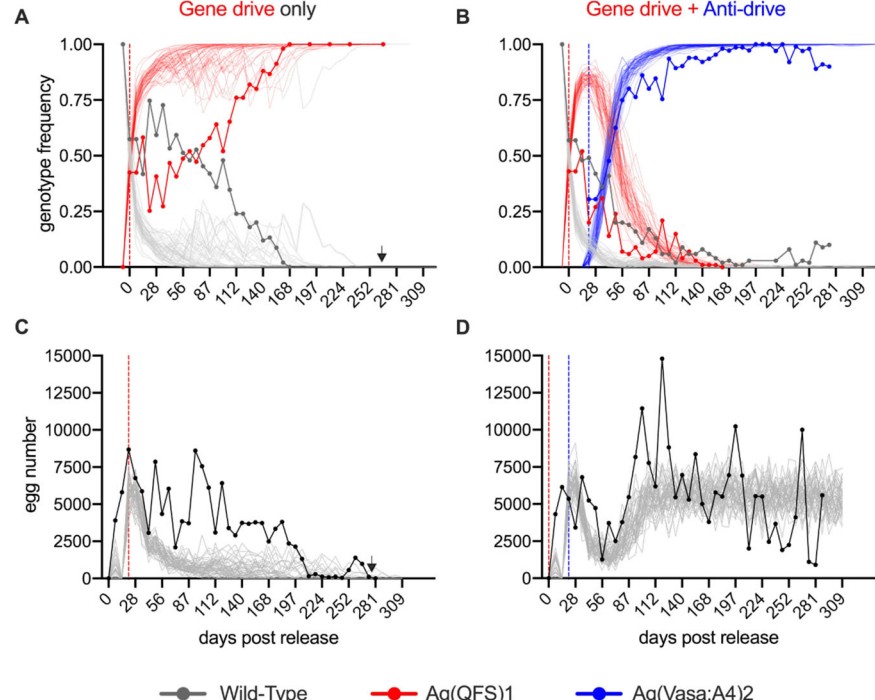

**Fig. 3 | Anti-CRISPR mosquitoes remove gene drive alleles in medium-sized cage after anti-drive release experiments.** In two medium-sized cages, a starting population of 400 wild-type *A. gambiae* mosquitoes (200 males and 200 females) were introduced; then, 150 mixed wild-type mosquitoes were introduced (restocked) every week in each population. After 2 weeks, in the cage named 'gene drive only', Ag(QFS)1 heterozygous males were released at 12.5% allelic frequency for 3 weeks on top of the wild-type restocking. For the cage called 'gene drive + anti-drive', following the 3 weeks of gene drive release, Ag(Vasa:A4)2 homozygous males were released at 15% allelic frequency, until the end of the experiments.

**A**, **B** Genotype frequency of Wild-type (dotted grey line), Ag(QFS)1 (dotted red line) and Ag(Vasa:A4)2 (dotted blue line) were monitored over time and compared to 50 stochastic simulations (light coloured lines). Dashed vertical lines indicate initial days of releases for the drive (red) and anti-drive (blue). **C**, **D** The total egg output (dotted black lines) was monitored for the two populations over time and compared to 50 stochastic simulations (grey lines). Fitness parameters for the modelling are provided in Supplementary Table 8. Arrows indicate time of population collapse.

same crossings were used to calculate the mating rate (Supplementary Fig. 2C); consistent with the above findings, trans-heterozygous females carrying the gene drive and anti-drive constructs showed an increased mating frequency (54.6%), compared to the females hemizygous for the gene drive (10.8%). A smaller increase in mating frequency has been observed also for males trans-heterozygous for the gene drive and the anti-drive construct (90.5%) compared to Ag(QFS) 1$^{+/-}$ males (73.3%).

## Ag(Vasa:A4)2 inhibits gene drive invasion in age-structured populations

Firstly, we investigated inhibition dynamics of the Ag(Vasa:A4)2 transgene in medium-sized cages, by setting up two overlapping age-structured wild-type populations, one named "gene drive only" and a second "gene drive + anti-drive", into which heterozygous Ag(QFS)1 gene drive males were released at 25% genotype frequency for three consecutive weeks (Figs. 2, 3). The genotype frequency of the

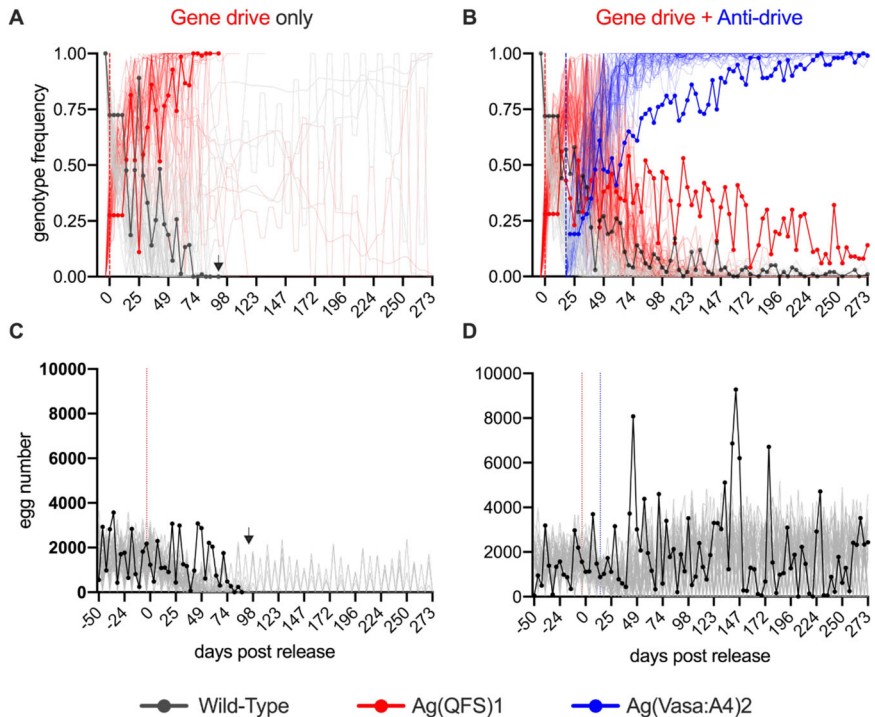

**Fig. 4 | Anti-CRISPR mosquitoes inhibit the spread of gene drive in large-sized cage after anti-drive release experiments.** Two age-structured wild-type populations were established in large-sized cages over a period of 70 days, then Ag(QFS) 1 heterozygous males were released twice a week at 25% allelic frequency for three consecutive weeks (day 0 in the graphs). For the cage named 'gene drive + anti-drive', following the gene drive release, Ag(Vasa:A4)2 homozygous males were released twice a week, at 30% allelic frequency, until the end of the experiment. **A**, **B** Genotype frequencies of wild-type (dotted grey line), Ag(QFS)1 (dotted red line) and Ag(Vasa:A4)2 (dotted blue line) were monitored over time and compared to 50 stochastic simulations (light coloured lines). Black arrows indicate the point at which no further eggs were recovered and the population was considered eliminated. Dashed vertical lines indicate initial days of releases for the drive (red) and anti-drive (blue). **C**, **D** Total egg output (dotted black line) collected after each bi-weekly feeding compared to 50 stochastic simulations (grey lines). Fitness parameters are provided in Supplementary table 9.

Ag(QFS)1 in the 'gene drive only' population oscillated at around 50% until day 98, then started increasing in frequency to reach fixation after 168 days (Fig. 3A), with a consequently decline in egg output, due to the increase of sterile homozygous Ag(QFS)1 females (Fig. 3C). The population completely collapsed, with no eggs produced, after 274 days. In the 'gene drive + anti-drive' population, the genotype frequency of the anti-drive Ag(Vasa:A4)2 rapidly increased, reaching 80% frequency after 63 days, and 90% frequency after 105 days (Fig. 3B). From day 197 until the end of the experiment, the frequency of the anti-drive fluctuated between 92% and 100%, while the Ag(QFS)1 frequency steadily declined until the gene drive was completely eliminated from the population at day 180 (Fig. 3B).

Subsequently, we assessed the anti-drive dynamics of Ag(Vasa:A4) 2 in large-sized cages, that could reveal different fitness costs compared to medium-sized cages, starting from two stabilized age-structured wild-type populations ('gene drive only' and 'gene drive + anti-drive'). As expected, Ag(QFS)1 frequency spread rapidly in the 'gene drive' only cage, reaching maximum frequency at day 70 and rendering the population completely sterile from day 88 leading to population collapse (when no more eggs were produced) (Fig. 4A, C). In the 'gene drive + anti-drive' cage, the genotype frequency of the Ag(QFS)1 did not exceed 56% (day 14), then progressively declined (as the anti-drive frequency increased), oscillating between 20% and 5% in the last 60 days of the experiment (Fig. 4B). The frequency of the anti-drive individuals increased from 19% at the time of release, up to 70% after 60 days, and above 88% from day 140 onwards. This data further support that the presence of the anti-drive in the population inhibits the spread of the gene drive and prevents population collapse, and that increasing genotype frequency of the anti-drive can maintain the gene drive frequency at low levels.

We performed pooled amplicon sequencing of the gRNA target site on samples collected at early and late time points after the gene drive release in the cages to monitor for gene drive resistance. The frequency of all indels observed around the target site did not exceed 0.3% of reads, with no increase in frequency over time (Supplementary Tables 5 and 6), providing evidence that functionally resistant alleles were not selected.

## Stochastic mathematical modelling of population dynamics for drive and anti-drive releases

We generated an agent-based stochastic model with overlapping generations mimicking the release scenarios performed in the medium- and large-size cages. The model used experimental and assumed fitness parameters for gene drive and anti-drive, to determine the relative fitness for each genotype (Figs. 3 and 4, and Supplementary Table 8 and 9). In the 'gene drive only' medium-sized population, we observed a slower Ag(QFS)1 spread compared to the model prediction (and consequently a slower reduction of wild-type individuals), although collapse time of the population falls well within the predicted range (Fig. 3A, Supplementary Fig. 5). Likewise, we observed a slower reduction of egg output compared to the model simulations (Fig. 3C). The observed data for the 'gene drive + anti-drive' population fit well with the modelling simulations (Fig. 3B) as well as the egg output, with the exception of the initial peak of gene drive frequency, even though the model does not capture the observed high stochasticity over time (Fig. 3D).

Simulations of population dynamics in large-sized cages closely correspond to the 'gene drive only' data in terms of Ag(QFS)1 spread, wild-type reduction and egg output dynamics (Fig. 4A, C, Supplementary Fig. 5). However, the model predicts the gene drive to persist

for long before reaching fixation in some simulations, and in few occurrences to disappear from the populations. The model overestimates the impact of the anti-drive releases on the population dynamics, predicting a faster increase of the anti-drive and complete removal of the gene drive genotype from the large cage population within the timeframe of the experiment (Fig. 4B, D).

Parameters such as the probability for any female to mate and lay eggs for age-structured populations in large cages is challenging to accurately measure. We therefore simulated a range of different scenarios with distinct overall fitness (Supplementary Figs. 6 and 7) by modulating the relative mating probability of the Ag(Vas a:A4)2 males (assuming same costs for heterozygous or homozygous individuals), which is a parameter that largely impacts the outcome of the dynamics (egg laying and the number of eggs are strictly dependent on mating). Because of the synergistic effects on mosquito fitness observed by the interaction of drive and anti-drive alleles, the model relieved such cost in the presence of the drive allele (i.e., trans-heterozygous). The $R^2$ values of the goodness of fit comparing the model-predicted to observed data showed that a mating probability of 20% best captures the dynamics of the genotypes in the large cage population ($R^2 = 0.815$, Supplementary Table 10 and Supplementary Fig. 8) and the trajectories of genotypes are well captured by the model output (Supplementary Fig. 7).

### Predicted impact of anti-drive release in simulated mosquito populations

Based on the best fitted parameters, we simulated different potential release scenarios to predict the impact of the Ag(Vasa:A4)2 on a population previously invaded by the Ag(QFS)1 gene drive with regards to time required for gene drive elimination (Ag(QFS)1 genotype frequency declining below 1% or 5%). We correlated populations of different sizes (from 500 to 5000 individuals, following the same biweekly restocking cycle of the experimental design) to gene drive frequencies in the population that would trigger the anti-drive release (Fig. 5). Every simulation for the medium-sized cages showed a fast

reduction of the drive under the 1% frequency in about 200–400 days or under 5% in 180–300 days, with little variation among population sizes (Fig. 5, left panels). The ability to eliminate the gene drive in large cage settings is instead strongly dependent on the population size and the initial frequency of the gene drive at the time of the anti-drive intervention. We observed that the larger the population size the longer is the time required to eliminate the gene drive and, interestingly, the intervention is more efficient if the anti-drive is released when the drive is already largely spread in the population (anti-drive release threshold of 0.75) (Fig. 5, right panels). By contrast, if the frequency of the gene drive is low (i.e. 0.15) the two alleles persist longer in the population (Fig. 5, right panels). This was confirmed when we modelled the frequency of the gene drive after 1500 days (Supplementary Fig. 9), and we observed that the number of simulations in which the gene drive is not eliminated increases with increasing population size and decreases when gene drive frequencies at the time of anti-drive release are higher.

## Discussion

In this study, we demonstrated the efficacy of a AcrIIA4-based anti-drive transgenic line to counteract a *doublesex*-targeting gene drive strain in a complex near-natural environment. The anti-drive construct blocks efficiently the spread of the gene drive in age-structured *An. gambiae* populations even if mosquitoes are exposed to behavioural and ecophysiological challenging conditions, preventing the collapse of the population.

We consistently observed complete inhibition of homing when individuals carried both anti-drive and gene drive transgenes, contrasting the strong supermendelian transmission rate of the gene drive. Moreover, confirming previous observations[21], the presence of the anti-drive construct partially restored the reduced fecundity and mating ability of the heterozygote Ag(QFS)1 females, most likely due to the AcrIIA4 inhibiting the somatic activity of Cas9 in the embryo after fertilisation[5,7]. We conclude that estimating fitness parameters of individual genotypes independently could lead to misinterpretation of

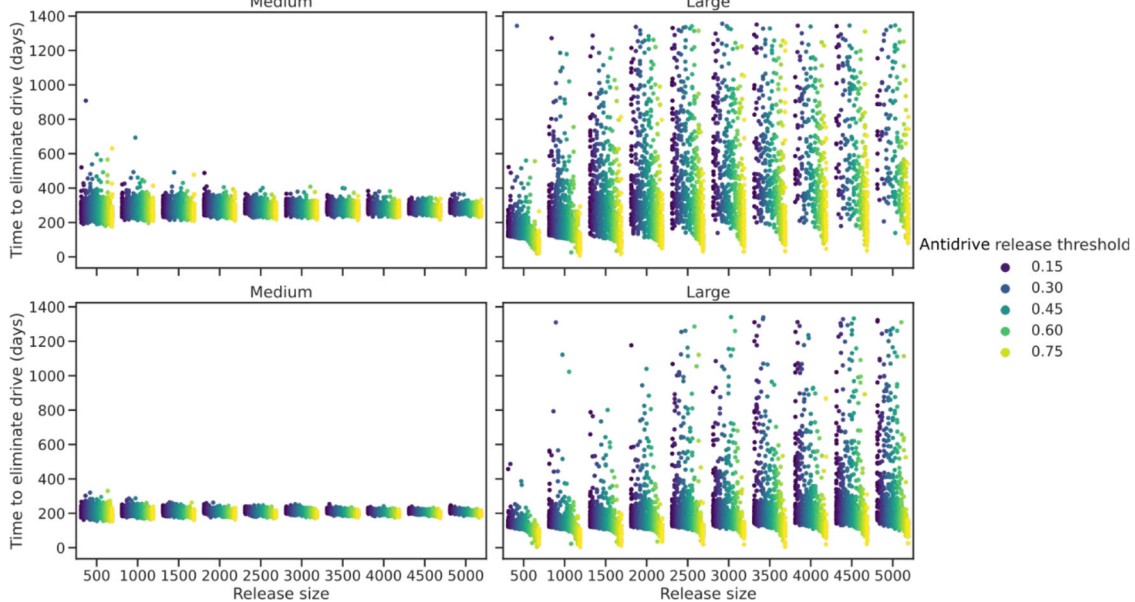

**Fig. 5 | Estimated time of gene drive removal from a mosquito population in function of various anti-drive release scenarios.** Fifty stochastic simulations of the time required for the elimination of the gene drive from the population after the introduction of the anti-drive (at 30% frequency), as a function of restocking release size (i.e. population size) and gene drive frequency in the population, for both medium-sized (left hand panels) and large-sized (right hand panels) cages. We define gene drive elimination when the genotype frequency of the drive is

maintained below the threshold of 1% (top panels) or 5% (bottom panels). The 'anti-drive release threshold' (different coloured dots) indicates the frequency of the gene drive in the population that triggers the release of the anti-drive in the model. Time (in days) is measured from the release of the anti-drive, and each dot identifies a simulation for each population size and frequency threshold pair. Data are plotted only for those simulations in which the gene drive frequency declined below the 1% or 5% threshold within the time limit.

the effective fitness when alleles co-occur in one genotype and the synergistic effect of gene drive and anti-drive constructs needs to be evaluated both separately and in combination. This is extremely important not only in the context of assessing the efficacy of a potential intervention, but also for accurate parameter estimation when assessing potential risks and/or benefits of releasing transgenic mosquitoes.

When investigating life-history traits in medium- and large-sized cages, we confirmed previous studies[5,31] that showed that adult longevity largely differs between the two experimental settings, reflecting the increased biological challenges induced by large cages. Although other fertility parameters could be inferred experimentally for the different genotypes, our mathematical modelling suggested that a priori parametrization does not capture the complexity of either the genotype interactions or the ecological complexity due to the environmental conditions. This is especially true for large cages where retrospective inference of specific life-history parameters from experimental data was required (for instance to estimate the egg laying probability of the wild-type population).

Large cage population studies additionally highlight the values of testing genetically modified mosquito strains in challenging ecological and behavioural conditions before field releases, following also stepwise approach recommendations[9,32]. We observed differences in the dynamics of the Ag(QFS)1 gene drive spread between the medium- and large-sized control cages, with a fast spread and collapse in the large cage and a slower, although steady, invasion in the medium-sized cage. The time required for Ag(QFS)1-induced population collapse in our experiments was successfully predicted by our mathematical model (Supplementary Fig. 5). The modelling estimates and the experimental validation confirms that the different parametrization due to the environment (i.e. large versus medium cage) has a much stronger impact on gene drive spread than the number of mosquitoes in the population (at least within the tested range).

By contrast, when it comes to the interactions between different transgenic strains with unlinked genetic modifications (in this case gene drive and anti-drive), the challenges for predicting accurately the population dynamics arise, especially in assessing all possible combinations and synergies or competitions. Indeed, the dynamics of drive inhibition in large cages indicated a lower efficacy in removing the drive from the population compared to medium-size cages (Fig. 3) where all simulations predict complete elimination of the gene drive. Again, the different life-history features of the mosquitoes in the large cages, where wild-type, gene drive and anti-drive genotypes interact, seem to have a stronger impact than that of the population size or the frequency of the gene drive at the time of the anti-drive intervention, as also suggested by the model (Fig. 5 and Supplementary Fig. 9). The low mating rate, egg laying probability and reduced adult survival observed in large cages, as a result of the more intricate ecological conditions, lead to an equilibrium where the gene drive persists at low frequency for long periods of time. The probability for this outcome increases with the population size, where stochastic effects are mitigated, and decreases with the initial rate of gene drive in the population. This is likely due to the lower probability for an anti-drive mosquito to encounter and mate with a gene drive mosquito, and therefore produce trans-heterozygote progeny in which the gene drive will be blocked and eventually eliminated from the population. These data would suggest that an anti-drive intervention would be more efficient if the anti-drive is released when the drive is already largely spreading in the population.

One could argue that the reduction in frequency of the gene drive alleles is attributed to genetic resistance being generated and positively selected at the gene drive target site. However, we showed no significant increase in the rate of mutated alleles at the gene drive target site, indicating no end-joining resistance selection, consistent with protein-based inhibition of the Cas9[21].

Our initial model based on experimental estimate of life-history traits showed an underestimation of gene drive performance and overestimation of the anti-drive fitness in large cages[5,31,33]. Posterior estimation of various fertility costs allowed however to estimate with a good fit the drive inhibition efficacy of the Ag(Vasa:A4)2 mosquitoes. Our modelling data suggest that mating rate is one of the parameters that largely impacts the dynamics of gene drive spread and removal (Supplementary Fig. 6 and 7). An effect that our model did not consider is the additional number of adult males in the cages as a result of the continuous release of anti-drive individuals that could have increased the mating of females and therefore contributed to the population rebound. This effect is also enforced by the introduction of wild-type alleles at the gene drive target locus by the continuous release of anti-drive mosquitoes, which reduces the absolute frequency of gene drive chromosomes every generation, counterbalancing the driving force.

Potential improvement of AcrII-based strategies for gene drive removal would be to genetically link the anti-drive transgene with the wild-type locus of the gene drive target (in this case the wild-type *dsx* allele), without impairing the locus functionality. In such a way, the wild-type target allele will be 'protected' by the anti-drive and function as a resistant allele, positively selected against the gene drive.

Our modelling outcome also showed some limitations to completely capture the biological complexity of genotype dynamics in different environmental settings, and the limit to simulate the high stochasticity we observed experimentally.

This study is the first successful test of anti-drive approaches in large cages that mimic behaviourally and ecophysiologically complex conditions, that have great potential utility at counteracting the spread of very effective population suppression gene drive. In the future, additional experimental validation, and modelling prediction to simulate release scenarios for the potential field use of such technology would be needed. For instance, in the context of field releases, it would be valuable to model gene drive spread and inhibition dynamics in a spatial manner, considering the effect of migration from neighbouring populations, larger population sizes, or geographical isolation[34,35].

## Methods

### Ethics
The research described in this study complies to all relevant ethical and legal requirements, and international standards for containment of genetically modified organism. All animal work was conducted according to the Italian regulations at Polo GGB and according to UK Home Office Regulations at Imperial College London.

### Containment and maintenance of mosquito strains
Three *Anopheles gambiae* mosquito strains were employed, the wild-type G3 strain (MRA-112), Q-driving Female Sterility strain, Ag(QFS)1[7] and a strain with the anti-CRISPR vasa2:AcrIIA4 construct as described in[21] (Supplementary Fig. 1A), integrated into a different genomic location via piggyBac mediated integration, here referred to as Ag(Vasa:A4) 2. This strain harbours the *Listeria monocytogenes* AcrIIA4 open reading frame, expressed under the *An. gambiae vasa* promoter and terminator elements and the 3xP3::eGFP fluorescent marker. *An. gambiae* mosquito strains were kept in a purpose-built Arthropod Containment Level 2 plus laboratory at Polo d'Innovazione di Genomica, Genetica e Biologia (Polo GGB), Genetics & Ecology Research Centre Terni, Italy as well as in an insectary that is compliant with Arthropod Containment Guidelines Level 2 at Imperial College London, UK[36]. Mosquitoes were held in cubical cages of 17.5 cm × 17.5 cm × 17.5 cm (BugDorm-4) as described in[37] at 28 °C and 80% relative humidity with a 12:12 h L:D photoperiod with 1 hr dawn and dusk simulation. Larvae were reared in plastic trays (253 × 353 x 81 mm) at a density of 200 individuals in 400 ml deionized water with sea salt at a concentration of 0.3 g/L and 5 mL of 2% w/v larval

diet. A Complex Object Parametric Analyzer and Sorter (COPAS, Union Biometrica, Boston, USA) was used to screen for the fluorescent markers.

## Plasmid construction

The *L. monocytogenes* AcrIIA4-coding sequence followed by an NLS at the N-terminus side, under the control of the vasa2 promoter[29], was amplified from C77 plasmid[21] using primers containing overhangs for Gibson assembly (RG964–RG969; Supplementary Table 7). A plasmid backbone containing the piggyBac inverted repeats and two φC31 attP recombination sites, as well as a fragment containing eGFP marker under the control of the 3xP3 promoter using primers also adapted for Gibson assembly (RG970–RG971 and RG968–RG967, respectively; Supplementary Table 7). The final plasmid was named C119 (Genbank accession code PRJEB61434) and was assembled using the standard Gibson assembly protocol[38].

## Generation of the Ag(Vasa:A4)2 strain

The G3 strain embryos were microinjected, as described previously[39]. The injected mix contained 50 ng/μL of the vasa2:AcrIIA4 construct and 400 ng/μL of a helper plasmid expressing the piggyBac transposase under the control of the *vasa2* promoter. The G$_0$ larvae with transient expression of the eGFP marker were crossed to wild-type mosquitoes to obtain transgenic individuals that were founders of the Ag(Vasa:A4)2 strain. The expression of fluorescent markers was analysed on a Nikon inverted microscope (Eclipse TE200). The Ag(Vasa:A4)2 strain individuals were crossed to individuals of the gene drive line Ag(QFS)1+/-. The trans-heterozygote offspring were crossed to an equal number of wild-type mosquitoes and the resulting progeny was counted and screened for inheritance of the gene drive (RFP positive) and the anti-drive (GFP positive) constructs. The Ag(Vasa:A4)2 strain was selected based on the mendelian inheritance pattern (Fig. 1C) and the rate of gene drive inhibition and the larval output (Supplementary Fig. 1B). Moreover, an inverse PCR (Supplementary Table 1), as previously described[40], on the strains selected were performed, to determine the integration locus of the anti-drive construct. Targeted nanopore sequencing with Cas9-guided adaptor ligation as well as whole genome nanopore sequencing, was used to determine the specific genomic location of the selected transgenic line, as described previously[41]. Specifically, for the targeted nanopore sequencing, high molecular weight (HMW) gDNA from -160 male and female transgenic individuals was extracted using an optimized HMW extraction protocol alongside QIAGEN Genomic-tip 20/G (cat#10223) and Genomic DNA Buffer Set (cat#19060). gRNA probes were designed using CHOPCHOP (sequences provided in Supplementary Table 7) and synthesized using synthetic CRISPR RNA (crRNA) and trans-activating crRNAs (tracrRNAs) to assemble a duplex. The resulting reads were mapped against a hybrid AgamP4-Ag(Vasa:A4)2 reference genome, in which the sequence of the Ag(Vasa:A4)2 transgene is appended to the latest AgamP4 genome file. BLASTn analysis of the reads aligning to the construct sequence was used to identify the insertion locus of the construct. Raw sequencing read are available in the EBI-ENA database (accession codes PRJEB61434, https://www.ncbi.nlm.nih.gov/bioproject/?term=PRJEB61434).

## Single deposition phenotypic assays

Transgenic males and females carrying either one copy (i.e. heterozygous) of the drive (RFP positive), one copy of the anti-drive (GFP positive), or both constructs (RFP and GFP positive) were crossed to wild-type independently for 5 days, blood-fed, and allowed to lay individually. Likewise, individuals homozygous and heterozygous for the Ag(Vasa:A4)2 insertion were crossed to wild-type and used for the same type of assay. For each genotype tested, 30 or 50 male or female adults were crossed to an equal number of wild-type counterpart. Eggs and larvae laid by each female were counted, and the inheritance of gene drive and anti-drive transgenes was scored through detecting the

expression of the linked RFP or GFP fluorescent markers respectively. Females that produced less than ten larvae or failed to produce progeny (with no evidence of sperm in their spermatheca) were excluded from the analysis. All the statistical analysis were performed using the Kruskal–Wallis's test using GraphPad Prism 9.

## Measuring life-history parameters

Life-history parameters were measured for Ag(Vasa:A4)2 and wild-type G3 in small cages (BugDorm-4) as described in Hammond et al.[5] assessing egg deposition, hatching rate, larval and pupal mortality, time of pupation and mating competition. Adult mortality assessment was conducted in both medium and large cages. To determine eggs number and hatching rate *en masse*, three replicate crosses were performed with 150 females and 120 males of the following genotypes: Ag(Vasa:A4)2+/- males with wild-type females; Ag(Vasa:A4)2+/- females to wild-type males; Ag(Vasa:A4)2+/+ males to Ag(Vasa:A4)2+/+ females; and wild type males to females. Females were blood-fed after 4 days, and the egg progeny was counted using EggCounter v1.0 software[42]. The hatching rate was estimated 3 days post oviposition, visually checking 200 eggs under a stereomicroscope (Stereo Microscope M60, Leica Microsystems, Germany). Time of pupation, larval and pupal mortality were evaluated by rearing three trays of 200 larvae/tray and counting/sexing the number of surviving pupae, in triplicate. Statistical analysis of the larval, pupal, and aquatic survivals (larval survival multiplied by pupal survival) was calculated using Kruskal–Wallis's test on the average of three replicates.

Mating competitiveness of Ag(Vasa:A4)2+/-, Ag(Vasa:A4)2+/+, and wild-type males was assessed in small cages, by placing 100 virgin 2-days old males with 100 2-days old virgin wild-type females, in triplicate. After 4–5 days, females were collected, and mating status was measured by the presence of sperm in the dissected spermatheca. Statistical differences were evaluated by Kruskal–Wallis's test. Sex-specific adult survival of wild-type, Ag(Vasa:A4)2+/- and Ag(Vasa:A4)2+/+ was performed in small and large cages. For small cages, 100 pupae were inserted in each cage divided by genotype and sex. Adult survival assay was performed in triplicate and calculated by the daily collection of dead mosquitoes. Adult survival in large cages was performed in duplicate, by monitoring the daily survival of 100 males and 100 females together in the same cage. Daily survival curves and statistical differences between genotypes and sexes were calculated using GraphPad Prism 9 and Kaplan-Meier test respectively.

## Large cage environment

In preparation for an anti-drive release study in large cages, overlapping generation populations of wild-type *An. gambiae* mosquitoes were maintained in a climatic chamber with two 6.4 m³ cages of 2.9 m x 0.96 m x 2.30 m (length, width, and height), with a temperature of 28 °C ± 0.5 °C and relative humidity of 80% ±5%. Two sets of three LEDs (3000, 4000, and 6500 K correlated colour temperatures) were used to illuminate the climatic chamber, controlled by Winkratos software (ANGELANTONI Industries S.p.A., Massa Martana, Italy), permitting to simulate of the dawn, dusk, and sunset. Full light conditions were adjusted to last 11 h and 15 min. The dusk lasted for 1 hr, to encourage the swarming behaviour of male mosquitoes; dawn lasted for 45 min from dark (11 h) to full light. The setting of the cages resembled the one described by[5]. In each cage, terracotta bricks resting shelter were placed and kept moistened with a soaked sponge; mosquitoes were allowed to feed on 10% sucrose and 0.1% methylparaben solution and blood-fed bi-weekly via Hemotek membrane feeder, using sterile cow blood (Allevamento blood di Ricci Chiara, Teramo, Italy). Two Petri dishes of 12 cm diameter with a wet filter paper strip were introduced into each cage to allow egg deposition 2 days after a blood meal. A black squared plastic marker (50 cm of side) was placed on the white floor to serve as a visual marker to stimulate swarming behaviour. At the front of each cage, two openings

allow the introduction of mosquito pupae to repopulate the cage, the introduction and collection of the egg dishes, the sugar feeders and the Hemotek feeders without any experimenter entering the cage for the whole duration of the experiment.

## Establishment, maintenance, and monitoring of age-structured large cage (ASL) populations

Two stable age-structured large (ASL) cage populations of *An. gambiae* (G3 strain) were established to investigate the potential of the Ag(Vasa:A4)2 anti-drive to inhibit the spread of the Ag(QFS)1 gene drive. The mean adult population size was estimated based on the mean adult daily survival (Kaplan-Meier estimate) and the biweekly restocking size (Data Source). Based on the weekly release of 1200 pupae (600 twice a week), we estimated a mean adult population size of 912 mosquitoes. 600 wild-type pupae (300 males and 300 females) were introduced twice a week (1200 total per week) in each large cage for 70 days (establishment) until the populations were producing enough eggs to self-maintain. Then, the only progeny of these populations was used to repopulate the cage (restocking) for 56 days (pre-release, 126 days total), with the introduction of wild-type mosquitoes reared separately when progeny numbers were too low. The ASL populations were defined as stabilized after 14 days in which the number of eggs produced from each cage was sufficient to restock the population. Blood feeding was performed in the morning for 5 h; eggs were collected 2 days later, counted, hatched in a tray, and reared within the same climatic chamber. Cage populations were restocked twice per week from a random cohort of 600 pupae at the peak of pupation, which were manually screened for sex. Experimental design graphics were created with BioRender.com

## Anti-drive release experiment in large-sized caged populations of overlapping generations

We have tested the capacity of the anti-drive Ag(Vasa:A4)2 to inhibit and counteract a very effective population suppressive Ag(QFS)1 gene drive strain in ASL populations of *An. gambiae* in large-sized cages. The design of the experiment and the release frequencies were chosen to validate the ability of the anti-drive to decline or even eliminate the gene drive from the populations in an experimentally reasonable timeframe. Based on preliminary theoretical modelling (Supplementary Fig. 10), we considered (1) heterozygote versus homozygote anti-drive males; (2) variable genotype frequencies of the anti-drive males releases and (3) simulated anti-drive fitness cost to estimate population dynamics. The model predicted that to remove the gene drive from the population a continuous release of homozygotes males was more efficient than releases of heterozygous, while a 50% releases did not substantially improved over a genotype frequency of 30% on top of the established populations, timewise, therefore 30% release was preferred. The allelic frequencies of each transgenic were calculated on the estimated mean adult pre-release population size of 912 adults (established previously on the mean adult survival). After stabilizing the two ASL wild-type populations, 228 heterozygous Ag(QFS)1 males were released twice a week, for three consecutive weeks (for a total of 456 pupae per week), on top of the 600 pupae used for restocking population (for a total of 1200 pupae per week), representing 25% allelic frequency of the estimated population. Then, the cage named 'gene drive only' was maintained by restocking randomly selected 600 pupae twice a week, while for the cage named 'gene drive + anti-drive', 136 homozygous Ag(Vasa:A4)2 males were continuously introduced every restocking on top of the randomly selected 600 pupae (272 total per week, representing 30% allelic frequency based on the estimated mean pre-release population of 912). Total egg output and hatching rate were recorded during the entire experiment. Larvae were reared at a density of 200 per tray, and 600 pupae were randomly selected and manually screened for sex and genotype by recording the presence of the RFP marker linked to Ag(QFS)1 and the GFP marker linked to

Ag(Vasa:A4)2 to evaluate the relative frequency of drive and anti-drive mosquitoes. Triplicate samples of up to 200 exuviae were collected weekly from the restocking population throughout the experiment and stored in absolute ethanol at −20 °C for subsequent molecular analysis.

## Anti-drive release experiment in medium-sized caged populations of overlapping generations

Two age-structured populations were established in medium-sized cages (30 × 30 x 30 cm) to mimic a similar release scenario of the large-sized cages. Based on the adult survival in medium-sized cages, the maintenance of the overlapping-generation population was adjusted to perform a single blood-feeding and a single restocking per week. The set-up of the populations was as follows: 400 pupae (200 males and 200 females) were introduced in the two medium-sized cages as a starting point; then, each week, 150 randomly selected pupae were introduced to maintain a mean adult population calculated as 425 mosquitoes based on adult mortality (Data Source). Subsequently, 3-week releases of 111 heterozygous Ag(QFS)1 males were released in both cages once a week (12.5% allelic frequency). In the second cage only, 66 homozygous Ag(Vasa:A4)2 males (15% allelic frequency) were introduced every restocking, on top of the 150 randomly selected pupae until the gene drive individuals were completely removed (day 274). After that point, no anti-drive mosquitoes were released, and the population was self-maintained and monitored. Egg output and hatching rate were recorded for every feeding. Larvae were reared at a density of 200 per tray. Transgenic frequency and sex ratio were recorded by a manual screening of randomly selected 150 pupae every week.

## Modelling

We modelled population and genotype dynamics using an agent-based stochastic model with overlapping generations. Shortly, in each run of the model, we considered an environment containing adult mosquitoes, which are allowed to mate randomly at specified times ("release day") and produce eggs. Only a predefined number ("release size") of the resulting pupae is reintroduced in the environment at one of the release days after they have matured. Each individual's life history parameters, such as probability to develop or adult longevity are drawn from empirically derived distributions, which depend on the adult genotype. In the model, we are considering two loci, locus 1 with three different alleles: W (wild-type), D (gene drive), or R (resistance); and locus 2 with two alleles: W (wild-type) or A (anti-drive). The genotype at locus 1 determines whether an individual is sterile: D/D, D/R, and R/R individuals can mate but do not produce eggs. The genotype at locus 2 determines whether homing is blocked in case the individual is heterozygous for the gene drive at locus 1. The following genotypes can block homing: W/A and A/A. The complete genotype was used to modulate several life history parameters such as mating probability, egg deposition probability, maturation probability, and drive/anti-drive efficiencies, using both empirically measured point estimates and reasonable assumptions. We assumed that females mate and lay eggs only once in their lifetime. We modelled three distinct base scenarios: an environment containing only wild-type individuals, one where only heterozygous gene drive males were released on top of the wild-type population for 3 weeks, and one in which homozygous anti-drive males were released continuously after the release of gene drive males. Each of the 50 simulations for each scenario was run for up to 500 days, with 0.1 days increments; after each time increase, all individuals' age was increased and those deceased or not developing into adults were removed. A run was stopped if the environment contained no individuals. We additionally modelled several scenarios, each with different permutations in the parameters used; broadly speaking these can be divided into 3 categories: different cage sizes, different release sizes and different fitness effects of the anti-drive. For the first category

we modelled two environments, large-sized cage with 600 pupae released twice a week, and medium-sized cage with 150 pupae released per week. For the first one we used a fitness estimate of -0.0478, indicating the proportion of females capable of both mating and laying eggs, while for the second the fitness was fixed at 0.66; the fitness parameter was estimated by running a parameter sweep on both mating and egg deposition probability and selecting the parameter pairs with the highest coefficient of determination ($R^2$) with the empirically measured number of eggs in a wild-type only cage (pre-release). Adult's lifespan was also set to be shorter in the larger environment compared to the smaller one, using a Weibull distribution fit on empirical data. This reflects the empirical observation that mosquito adult longevity is shorter in large cages environments. For the second category we modelled release sizes ranging from 500 to 5000 pupae for each restocking. For the last scenario, we modelled varying effects of the anti-drive on fitness; for individuals that carried the anti-drive construct we imposed a fitness modifier ranging between 0.01 (strong effect) to 1 (no effect). The modifier was applied to the base mating probability, regardless of the genotype at locus 1. Base parameters and genotype specific modifiers for both environments are reported in the Supplementary Table 9 and 10. The model is implemented in the python programming language, using the numpy[43], scipy[44] and pandas[45] libraries. The model code and parameter files for each presented run are available on github at https://github.com/khalillab/large-cage and on *Zenodo* at https://zenodo.org/records/10404587 (https://doi.org/10.5281/zenodo.10404587), together with a snakemake[46] pipeline for reproducibility.

#### Amplicon sequencing and analysis

Analysis of gene drive target site resistance by deep-amplicon sequencing was performed as previously described in[7], with the difference of using exuviae from the restocked samples instead of L1 larvae. Samples were collected in duplicate from both experimental large-sized cages at 4, 21, 49, 84, 105, 203, 231 and 273 days post-release, and genomic DNA extracted in pool (using DNeasy Blood & Tissue Kit, Qiagen). To investigate presence of mutations at the target site of Ag(QFS)1, DNA from pooled samples was PCR amplified using primers 4050-Illumina-F TCGTCGGCAGCGTCAGATGTGTATAAGAGA CAGACTTATCGGCATCAGTTGCG and 4050-Illumina-R GTCTCGT GGGCTCGGAGATGTGTATAAGAGACAGGTGAATTCCGTCAGCCAGCA (Illumina adaptors are underlined). The libraries were prepared following the Illumina 16S Metagenomic Sequencing Library Preparation protocol and the Nextera XT index kit. The raw sequencing data were analysed using *CRISPResso2*[47], setting the minimum average read quality score (*phred33*) to 30 and a window of 20 bp surrounding the cleavage site was preferred to evaluate indels and substitutions as previously described[7]. Raw sequencing data are available at the EBI-ENA database (accession code PRJEB61434).

#### Statistics & Reproducibility

All statistical tests performed are indicated in the figure legends or in the methods section, as appropriate. No data were excluded from the analyses, unless stated. Information on the assays and the study design is provided to allow reproducibility of the results. The Investigators were not blinded to allocation during experiments and outcome assessment. Sample size for the analysis of fitness parameters is consistent with previous literature reporting similar phenotype assays.

#### Reporting summary

Further information on research design is available in the Nature Portfolio Reporting Summary linked to this article.

## Data availability

Raw-sequencing data generated in this study have been deposited in the EBI-ENA database under accession code PRJEB61434 for the targeted nanopore sequencing, the whole genome nanopore sequencing and for the amplicon sequencing, while the hybrid AgamP4-Ag(Vasa:A4)2 reference genome is provided as a fasta file in the source data. The C119 plasmid sequence has been deposited in the GenBank database under accession code PRJEB61434. Source data are provided with this paper.

## Code availability

The model code and parameter files for each presented run are available on github at https://github.com/khalillab/large-cage and on *Zenodo* at https://zenodo.org/records/10404587 (https://doi.org/10.5281/zenodo.10404587), together with a snakemake[46] pipeline for reproducibility.

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

## Acknowledgements

We thank Jarek Krzywinski for useful comments on the manuscript. This work was supported by the Defense Advanced Research Projects Agency (HR0011-17-2-0042) for this research to C.T., M.G., R.L.M., A.GT.S., R.M., A.S.K., A.S., R.G., and A.C. The views, opinions and/or findings expressed should not be interpreted as representing the official views or policies of the Department of Defense or the U.S. Government. The salary of R.DA., A.T. and S.G. were supported by the Bill & Melinda Gates Foundation (OPP1210755). M.G. was further supported by the Deutsche Forschungsgemeinschaft (DFG, German Research Foundation) under Germany's Excellence Strategy - EXC 2155 - project number 390874280. A.S.K. acknowledges funding from a Department of Defense Vannevar Bush Faculty Fellowship (N00014-20-1-2825).

## Author contributions

R.M., A.S., A.C., and R.G. conceptualized the work; R.D.A., C.T., A.S., and R.M designed the research; R.D.A., C.T., A.T., R.L.M., S.G., A.GT.S. performed the research; R.D.A., C.T., and A.S. analysed data; A.S.K and M.G. developed mathematical modelling. R.D.A, C.T., and A.S. wrote the manuscript with inputs from R.M. All authors reviewed the manuscript.

## Competing interests

Imperial College London and Polo GGB have patents pertaining to the use of anti-CRISPR constructs to counteract gene drives in arthropods (Inventors: A.C., R.G, R.D.A., C.T. and A.S.). A.C. is a founder of Biocentis, Ltd and has an equity interest in Biocentis Ltd. Other authors declare no competing interests.
