## [Peer Review File · Nature Communications]

Reviewers' Comments:

Reviewer #1:

Remarks to the Author:

The authors present a study demonstrating the development of an improved anti-CRISPR-based anti-drive mosquito strain and its efficacy in inhibiting gene drive strains in medium to large cages. The authors also simulated the dynamics of inhibition using a stochastic model. This work fortifies the existing evidence that anti-CRISPR-engineered strains could be useful in curtailing the proliferation of gene drive in *Anopheles* mosquitoes.

The impact of this study needs to be considered within the context of relevant research, particularly a similar paper by Taxiarchi et al., published in *Nature Communications* in 2021. This study uses the same anti-CRISPR molecule, AcrIIA4, and the major difference is its insertion into a different genomic locus. The authors argue that this modification offers advantages such as reduced fitness costs and improved survival rate to adulthood when in homozygosity. However, these benefits, which seemingly form the crux of this study's major contributions, lack sufficient characterization. As such, a more comprehensive evaluation of the benefits conferred by the new anti-drive strain is needed to establish the merit of these purported advancements.

In light of published works, it is not surprising that their improved anti-drive strain can effectively inhibit gene drive strains. However, it remains unclear how the improved strain performs compared to the previously published anti-drive strain. Since both strains utilize the same anti-CRISPR molecules, a direct comparison should be performed to ascertain the degree to which this new strain represents a significant advance in the gene drive field. At present, beyond a possible improvement in fitness cost, it is challenging to definitively declare this new anti-drive as a major advance.

It is also notable that several works have been published on using anti-CRISPR systems to inhibit genome editing or regulation in contexts beyond their native bacterial environment including mammalian cells and systems. Therefore, the authors should thoroughly reference these previous works to acknowledge the research that has paved the way for their current study.

Reviewer #2:

Remarks to the Author:

This is an interesting manuscript in which the authors create new insertions of an anti-crispr protein in *Anopheles* and follow its ability to block homing and population suppression by a doublesex homing drive.

This is a follow up to an earlier report in which the authors showed that the anti-crispr protein could prevent a population collapse driven by this same homing drive. However, fitness costs associated with the construct probably prevented it from driving to higher frequency and/or driving the homing drive to very low frequencies, or entirely out of the population.

To summarize, here the authors show that a new insertion of an anti-crispr protein, driven as previously, by the *vasa* regulatory sequences, has higher fitness in isolation. They also show that when released repeatedly in a challenging environment, it increases in frequency, while driving the HEG frequency down to low levels, and in some cases to extinction.

The results are important and the underlying data is overall clear. In the below I have one long question about how the experiments carried out are motivated by the accompanying text and relate to the earlier experiments. I cannot comment on the details of the parameterized modeling but have a few comments on the conclusions regarding the limitations of modeling.

1. In the earlier work the authors did a one-time addition of anti-crispr-bearing mosquitoes to a wildtype population in which a *dsx* homing drive was introduced at the same time. They could do a single release because this was a discrete generation drive experiment. The key result from this work was that the anti-crispr construct spread, while blocking the HEG from spreading to fixation,

thereby preventing population elimination. This result is important as it provides a very straightforward and unbreakable way of preventing unwanted HEG-driven population extinction. But, as the authors note in creating their new experimental design, the old setup does not provide the behavioral and environmental challenges that larger, age-structured populations would. Thus, these earlier experiments leave it unclear if the original anti-crispr construct would survive and thrive in a more realistic environment.

2. The drive experiments in the current manuscript use this more challenging behavioral and physical environment and a new anti-crispr element. This kind of experiment is important as it gets us closer to how drive and anti-drive elements would interact in the wild. However, the current experiments involve releases every generation, once started. It would help the reader visualize this key difference if Supplement Fig. 5, the schematic, was included in the main text. It is present in the written methods, but the text is not very clear and its a somewhat unusual design.

3. Related to the point about supplement Fig 5 above, I think the authors could spend a little more time motivating this choice of experimental design. The authors give a nice description of the new anti-crispr line and provide multiple lines of evidence that it has increased fitness as compared with the original line. And the use of large cages and age structure is great, as it provides a more realistic environment. At this point they might have repeated the drive experiment from the earlier paper (a single release, or rather three releases, as with the drive element introduction into the structured population) as a way of showing that the current element drove to higher frequency and led to a greater decrease in frequency of the homing cassette, due to its increased fitness. Or it might not have done so well, in these new more challenging conditions. But, they did not do this (and, to be clear, I am not suggesting that they go back and do it at this point).

I understand that the current experimental setup, with overlapping generations and age structure, provides a more rigorous and realistic test of fitness. But, its unclear what the motivation was for doing a new experiment (with continuous releases) that cannot be compared with the old one.

In addressing my confusion, and more generally in terms of providing a point of reference for readers, it may help to outline in the introduction something like the target product profile of what they feel they need to be able to achieve in the wild with the anti-crispr, and what the contexts are. This might help to motivate the specific experiment carried out.

A. The primary goal would seem to be to prevent population elimination, since this is what the HEG is designed to do. The earlier work achieved this goal with a single release, albeit leaving the HEG at a significant frequency in the population (we dont know if multiple releases would also have resulted in elimination). Since mutation of the anti-crispr construct to inactivity is selected against in the presence of the HEG (and the wildtype version is selected for), there is no real concern that the system will fail.

B. A secondary goal could be to actually eliminate the HEG. This seems to be the target for the some of the current experiments, but this choice of endpoint – this versus simple maintenance of a viable population – is never motivated in the text or made explicitly clear as a goal.

C. What is the context in which the anti-crispr is to be used? The release paradigm used in this manuscript works to drive the HEG to extinction or to very low levels, but what are the contexts in which you would be able to do large scale releases of an anti-crispr every generation (15-30% allele frequency) over the range within which the HEG is rapidly spreading? I understand that the HEG is what drives the anti-crispr into the population, but its not homing. It spreads due to death of those that do not carry it, much like a toxin-antidote system. So its a bit slow, and if a HEG is spreading through space the anti-crispr will always be chasing it far behind since selection for the anti-crispr is only strong when the HEG frequency is high. Are the releases in every generation meant to simply speed up something that would happen with a single release (in a bounded population), but just take a lot longer (and so not be compatible with the understandable need to publish in a timely manner)? Or is there another reason/context in which this makes sense? In short, its easy to understand the significance of experiments in which a single release stops a HEG from eliminating a population. Its less clear what to make of ones in which releases are carried out

continuously. An interesting related question is whether the model they arrive at predicts that a bolus of three releases (equivalent in some sense to a single release in the earlier discrete generation experiments) in this more challenging environment would work? Or does the model tell us we really need those continuous releases?

All of this (continuous releases) could make some sense in the context of thinking about releases in isolated environments like islands, where one might want to have the ability to go in and actually eliminate the HEG from the population at some point. I am not sure if this is where the authors are going with these experiments.

Finally, it might be useful to say a few words about how the HEG is eliminated from the population. It seems likely several forces are at work. It is lost in homozygous HEG females, as always (a recessive effect). Also, as the anti-crispr individuals are introduced and increase in frequency it begins to behave as a Mendelian allele. As this happens any dominant (but not so much recessive) fitness costs will weed it out of the population relatively (in evolutionary terms, which can be quite slow on human timescales) rapidly. In addition, the fact that the anti-crispr insects being introduced into the population every generation are wildtype at the HEG target locus serves to dilute the HEG-bearing chromosome out each generation. It is the anti-crispr protein that provides the condition under which other forces like homozygous sterility, fitness costs and dilution can do their work. But it might be worth noting the extent to which these other forces are contributing to ultimate elimination (for example, I am not sure what the dominant fitness costs of carrying the HEG are when it is forced to behave as a Mendelian allele).

To summarize, in the introduction the authors note that anti-crispr can be used to prevent spread in some contexts (left largely unspecified). The current manuscript shows that an anti-crispr can in fact be used to eliminate a HEG from the population, a significant achievement. What is missing, I think, is more context relating these results to earlier work, and contexts in which the strategies demonstrated in this manuscript (or other strategies) can be implemented. In short, I am just suggesting a bit more discussion and context creation to motivate the experiments carried out and results observed.

One general comment on the modeling discussion.

In the abstract the authors state that " A stochastic model predicted the experimentally-observed genotype dynamics in overlapping generations in medium- and large-sized cages and further demonstrated the effectiveness of anti-drive in different release and fitness scenarios. "

However, in the modelling section and discussion the authors talk at length about the ways in which the model failed to capture behaviour when based on measurements of individual parameters. Would it make sense to change the abstract to something that indicates both strengths and limitations of modelling in environments with many complex unknowns? One take away I had from their work is that their results showed how much modelling needed to be informed by the results of experiments.

Bruce Hay

Reviewer #3:

Remarks to the Author:

D'amato et al. have developed a gene drive inhibitor system based on the AcrIIA4 strain, which is an anti-CRISPR molecule approach. Although their system is building on a previous AcrIIA4 system, their strain improved life-history fitness costs compared to previous constructs. They also develop a stochastic agent-based model to evaluate release parameters and fitness cost characteristics that determine frequency of anti-drive, gene drive and wild type alleles in large and medium cages. Overall, this is a well written article and it is the first demonstration of anti-drive impacts in large cage trials. I have two specific comments that I would like addressed. One, I would like the authors to explain why there is a mismatch in model results compared to data from the cages - for example, with the time to establishment of populations varying between model and

actual cage trials. More specifically, I would like the authors to explain what could be missing in their model to result in this mismatch because it is very likely models will be used to determine releases in field settings and a mismatch at the cage trial level may only lead to more uncertainty in larger simulations. Second, could the authors comment on what happens when there are 'islands' of gene drive mosquitoes that can't be reached with anti-drive. Or are the authors proposing an optimal time of release (e.g. in the growth phase of gene drive spread) is all that's necessary to permanently inhibit gene drives in a population?

(Minor comment) Scale-invariant probabilistic loss function:

- Typo at the end of sentence, "...structured overlapping populations that involve more complex behaviour and ecologically conditions."
- Uncommon citation style for references in text. For example "as described in 5 assessing egg deposition," Line 189 on page 4.

Anti-CRISPR *Anopheles* mosquitoes inhibit gene drive spread under challenging behavioural conditions in large cages

Response to the reviewers' comments

Dear editor, dear reviewers,

we thank the reviewers for their extremely useful comments. We improved the introduction including a more complete reference to previous works on Acrs, including mammalian systems. We largely revised the discussion expanding on the context to use anti-drive and the motivation for the study described in the manuscript, as per reviewers' comments. We also improved the methods with a clearer description of the experimental design, and moved the supplementary figure 5 (the schematic of the experimental design) to the main text to help the reader follow the design description.

We grouped our response to the reviewers based on common comments, in the following, and a point-by-point to specific comments thereafter (in blue).

COMMON RESPONSES TO REVIEWER COMMENTS

Common response #1. Lack of comparison with the previously generated anti-drive strain.

The main focus of this study is the assessment of an anti-drive strain in complex ecological and behavioural conditions to validate its ability to inhibit gene drive spread in a contained environment. This follows a step-wise testing of GM strains under increasing complexity, following also international guidelines and recommendations (see for instance the WHO guidance report on GMM testing). We have not performed and would not want to focus on a direct comparison between different strains, as it falls outside the scope of our objectives. Accordingly, we have toned down the emphasis on the increased fitness exhibited by the newly developed strain (as the first strain failed to be maintained in homozygosity), while we focused more on the experiment's primary objective.

Common response #2. The main experimental objective and the rationale for the experimental design

The main objectives of the current study were the validation of an anti-drive strategy based on AcrII4 to inhibit and counteract a very effective population suppressive gene drive strain. After phenotype characterization, we validated the ability to stop the spread and revert the suppressive impact of a gene drive already invading in a population, in age-structured overlapping generations in medium and large-sized cages. The main focus was to challenge a validated anti-drive approach in complex ecological and behavioural conditions, following recommendation to test GM mosquitoes in increasing level of complexity in a step-wise approach pathway towards potential field release. In order to validate this approach in a logistically and experimentally reasonable timeframe we designed the experiment and the release based on preliminary theoretical modelling. A continuous release of homozygous anti-drive was predicted to eliminate the gene drive from the population more efficiently than the release of heterozygous individuals, while a 50% release did not have important improvement compared to a 30% release (in terms of timeline to replace the drive nor in terms of dynamics). Therefore 30% anti-drive release was chosen as a more realistic option, considering the number of released mosquitoes. Therefore, we first established a wild-type population, which was then invaded by a gene drive. The anti-drive strain was released on top of the established population

to simulate a plausible release scenario. A more detailed explanation of the experimental design and the rationale is now included in the methods section.

Common response #3. The limitation of the mathematical modelling

We highlighted in the discussion the limitation of the modelling to completely capture some of the biological complexity of age-structured populations, especially relative to the population stochasticity and the uncertainties to fully evaluate some fitness parameters. Therefore, the value of GM testing in large cages, and with complex population structure, is the ability to infer fitness parameters a posteriori which are fundamental to fully characterize GM strains and predict population dynamics (or to validate different modelling predictions). In this respect we have improved the discussion to better describe the limitations and the benefit of the approach we undertook. This is extremely relevant, since predicted population dynamics where the model is fed on parameters that are inferred by such behaviourally-challenging experiments (although still lab based) provide more informative predictions than parameters collected in completely artificial settings which can lead to under- or over-estimated mosquito fitness.

Common response #4. Considerations on a possible anti-drive intervention

While we agree with the reviewers' comment that it is extremely interesting discussing how anti-drive might be used in the field to control or recall gene drive spread, this would be a topic of a dedicated study. We in fact highlighted in the discussion that 'In the future, additional experimental validation, and modelling prediction to simulate release scenarios for the potential field use of such technology would be needed. For instance, in the context of field releases, it would be valuable to model gene drive spread and inhibition dynamics in a spatial manner, the effect of migration from neighbouring populations, larger population sizes, or geographical isolation'. The main scope of this study was to evaluate if the currently available anti-drive strain based on AcrA4 molecule can prevent spread of an efficient gene drive in challenging conditions, in a contained lab, as a step-wise development process. Once this step is validated, additional specific questions for potential field use need to be addressed, including better modelling predictions based on spatial information and different, more complex population dynamics. These could consider, for instance, population migration, larger population size, geographical isolations, local population collapse in pockets not reachable by the interventions, etc.

POINT-BY-POINT RESPONSES TO REVIEWER COMMENTS

Reviewer #1 (Remarks to the Author):

The authors present a study demonstrating the development of an improved anti-CRISPR-based anti-drive mosquito strain and its efficacy in inhibiting gene drive strains in medium to large cages. The authors also simulated the dynamics of inhibition using a stochastic model. This work fortifies the existing evidence that anti-CRISPR-engineered strains could be useful in curtailing the proliferation of gene drive in *Anopheles* mosquitoes.

The impact of this study needs to be considered within the context of relevant research, particularly a similar paper by Taxiarchi et al., published in *Nature Communications* in 2021. This study uses the

same anti-CRISPR molecule, AcrIIA4, and the major difference is its insertion into a different genomic locus. The authors argue that this modification offers advantages such as reduced fitness costs and improved survival rate to adulthood when in homozygosity. However, these benefits, which seemingly form the crux of this study's major contributions, lack sufficient characterization. As such, a more comprehensive evaluation of the benefits conferred by the new anti-drive strain is needed to establish the merit of these purported advancements. In light of published works, it is not surprising that their improved anti-drive strain can effectively inhibit gene drive strains. However, it remains unclear how the improved strain performs compared to the previously published anti-drive strain. Since both strains utilize the same anti-CRISPR molecules, a direct comparison should be performed to ascertain the degree to which this new strain represents a significant advance in the gene drive field. At present, beyond a possible improvement in fitness cost, it is challenging to definitively declare this new anti-drive as a major advance.

Thank you for this very important comment which showed us, that our main objective was not written up clearly enough. In the revised version, we clarified the main scope of this study and relieved the reader's expectations on the improved characteristic of the newly developed strain (lines 108-114). See also common responses # 1 and 2.

It is also notable that several works have been published on using anti-CRISPR systems to inhibit genome editing or regulation in contexts beyond their native bacterial environment including mammalian cells and systems. Therefore, the authors should thoroughly reference these previous works to acknowledge the research that has paved the way for their current study. As suggested, we incorporated additional references as appropriate (line 76-87).

Reviewer #2 (Remarks to the Author):

This is an interesting manuscript in which the authors create new insertions of an anti-crispr protein in anopheles and follow its ability to block homing and population suppression by a doublesex homing drive. This is a follow up to an earlier report in which the authors showed that the anti-crispr protein could prevent a population collapse driven by this same homing drive. However, fitness costs associated with the construct probably prevented it from driving to higher frequency and/or driving the homing drive to very low frequencies, or entirely out of the population. To summarize, here the authors show that a new insertion of an anti-crispr protein, driven as previously, by the vasa regulatory sequences, has higher fitness in isolation. They also show that when released repeatedly in a challenging environment, it increases in frequency, while driving the HEG frequency down to low levels, and in some cases to extinction. The results are important and the underlying data is overall clear.

Thank you for this overall positive evaluation!

In the below I have one long question about how the experiments carried out are motivated by the accompanying text and relate to the earlier experiments. I cannot comment on the details of the parameterized modeling but have a few comments on the conclusions regarding the limitations of modeling.

1. In the earlier work the authors did a one-time addition of anti-crispr-bearing mosquitoes to a wildtype population in which a dsx homing drive was introduced at the same time. They could do a single release because this was a discrete generation drive experiment. The key result from this work

was that the anti-crispr construct spread, while blocking the HEG from spreading to fixation, thereby preventing population elimination. This result is important as it provides a very straightforward and unbreakable way of preventing unwanted HEG-driven population extinction. But, as the authors note in creating their new experimental design, the old setup does not provide the behavioral and environmental challenges that larger, age-structured populations would. Thus, these earlier experiments leave it unclear if the original anti-crispr construct would survive and thrive in a more realistic environment.

Thank you for this important comment which showed us, that our major objective was not written up clearly enough! In the revised version, we clarified the main scope of this study and relieved the reader's expectations on the improved characteristic of the newly developed strain (lines 108-114). See also common responses # 1 and 2.

2. The drive experiments in the current manuscript use this more challenging behavioral and physical environment and a new anti-crispr element. This kind of experiment is important as it gets us closer to how drive and anti-drive elements would interact in the wild. However, the current experiments involve releases every generation, once started. It would help the reader visualize this key difference if Supplement Fig. 5, the schematic, was included in the main text. It is present in the written methods, but the text is not very clear and its a somewhat unusual design.

As suggested by the reviewer, we moved the suppl Fig 5 to the main text and expanded the methods to better describe the design and the rationale (lines 275-305).

3. Related to the point about supplement Fig 5 above, I think the authors could spend a little more time motivating this choice of experimental design. The authors give a nice description of the new anti-crispr line and provide multiple lines of evidence that it has increased fitness as compared with the original line. And the use of large cages and age structure is great, as it provides a more realistic environment. At this point they might have repeated the drive experiment from the earlier paper (a single release, or rather three releases, as with the drive element introduction into the structured population) as a way of showing that the current element drove to higher frequency and led to a greater decrease in frequency of the homing cassette, due to its increased fitness. Or it might not have done so well, in these new more challenging conditions. But, they did not do this (and, to be clear, I am not suggesting that they go back and do it at this point). I understand that the current experimental setup, with overlapping generations and age structure, provides a more rigorous and realistic test of fitness. But, its unclear what the motivation was for doing a new experiment (with continuous releases) that cannot be compared with the old one.

As per request, we expanded the methods section to better describe the experimental design and clarified in the introduction the rationale for this study, also considering the previous experiments. See also common responses # 1 and #2.

In addressing my confusion, and more generally in terms of providing a point of reference for readers, it may help to outline in the introduction something like the target product profile of what they feel they need to be able to achieve in the wild with the anti-crispr, and what the contexts are. This might help to motivate the specific experiment carried out.

In accordance, the introduction was updated to better clarify the aim of this study and the methods section was updated to better describe the experiment design and rationale (lines 88-90, 108-115, and 275-304). We kindly ask the reviewer to read our common responses #1, #2 and #3.

A. The primary goal would seem to be to prevent population elimination, since this is what the HEG is designed to do. The earlier work achieved this goal with a single release, albeit leaving the HEG at a significant frequency in the population (we don't know if multiple releases would also have resulted in elimination). Since mutation of the anti-crispr construct to inactivity is selected against in the presence of the HEG (and the wildtype version is selected for), there is no real concern that the system will fail.
B. A secondary goal could be to actually eliminate the HEG. This seems to be the target for some of the current experiments, but this choice of endpoint – this versus simple maintenance of a viable population – is never motivated in the text or made explicitly clear as a goal.

This consideration is now included in the text as follows: “The design of the experiment and the release frequencies were chosen to validate the ability of the anti-drive to decline or even eliminate the gene drive from the populations in an experimentally reasonable timeframe. Based on preliminary theoretical modelling (data not shown), we considered 1) single versus continuous release; 2) heterozygote versus homozygote anti-drive males; and 3) variable genotype frequency of the anti-drive males releases. The model predicted that to remove the gene drive from the population a continuous release of homozygotes males was more efficient than releases of heterozygotes, while a 50% release did not substantially improve over a genotype frequency of 30% on top of the established populations, timewise, therefore 30% release was preferred. The allelic frequencies of each transgenic were calculated on the estimated mean adult pre-release population size of 912 adults (established previously on the mean adult survival)” (lines 278-289).

C. What is the context in which the anti-crispr is to be used? The release paradigm used in this manuscript works to drive the HEG to extinction or to very low levels, but what are the contexts in which you would be able to do large scale releases of an anti-crispr every generation (15-30% allele frequency) over the range within which the HEG is rapidly spreading? I understand that the HEG is what drives the anti-crispr into the population, but it's not homing. It spreads due to death of those that do not carry it, much like a toxin-antidote system. So it's a bit slow, and if a HEG is spreading through space the anti-crispr will always be chasing it far behind since selection for the anti-crispr is only strong when the HEG frequency is high. Are the releases in every generation meant to simply speed up something that would happen with a single release (in a bounded population), but just take a lot longer (and so not be compatible with the understandable need to publish in a timely manner)? Or is there another reason/context in which this makes sense? In short, it's easy to understand the significance of experiments in which a single release stops a HEG from eliminating a population. It's less clear what to make of ones in which releases are carried out continuously. An interesting related question is whether the model they arrive at predicts that a bolus of three releases (equivalent in some sense to a single release in the earlier discrete generation experiments) in this more challenging environment would work? Or does the model tell us we really need those continuous releases? All of this (continuous releases) could make some sense in the context of thinking about releases in isolated environments like islands, where one might want to have the ability to go in and actually eliminate the HEG from the population at some point. I am not sure if this is where the authors are going with these experiments.

The rationale for choosing a continuous release of anti-drive on top of the population was to observe the reduction or elimination of the drive in a shorter timeframe than would have occurred with fewer releases, and to determine the differences observed in different experimental settings (medium vs large cages). Please read also our common responses #1, #2 and #4.

Finally, it might be useful to say a few words about how the HEG is eliminated from the population. It seems likely several forces are at work. It is lost in homozygous HEG females, as always (a recessive effect). Also, as the anti-crispr individuals are introduced and increase in frequency it begins to behave as a Mendelian allele. As this happens any dominant (but not so much recessive) fitness costs will weed it out of the population relatively (in evolutionary terms, which can be quite slow on human timescales) rapidly. In addition, the fact that the anti-crispr insects being introduced into the population every generation are wildtype at the HEG target locus serves to dilute the HEG-bearing chromosome out each generation. It is the anti-crispr protein that provides the condition under which other forces like homozygous sterility, fitness costs and dilution can do their work. But it might be worth noting the extent to which these other forces are contributing to ultimate elimination (for example, I am not sure what the dominant fitness costs of carrying the HEG are when it is forced to behave as a Mendelian allele).

This point raised by the reviewer is very interesting and pertinent. There are indeed many factors at play to impact the dynamics of drive invasion and anti-crispr control. To simplify, the relative fitness of each genotype, together with the homing rate (for the drive) and the homing inhibition (for the anti-drive) are the main factors which determine the overall dynamics (excluding here the contribution that drive-resistant alleles could play. We can exclude them here because we did not observe any mutations at frequencies which could have impacted the dynamics). There is indeed a 'dilution' effect of wt alleles at the drive target locus by the continuous release of anti-drive which could decrease (or halt) the spread of the drive. This could also happen for instance if there is a very large migration between connected populations. This dilution effect is larger the lower the relative fitness of the drive is, and it is counteracting the homing drive. What differs with the anti-drive is the outcome as the anti-drive increases in frequency, and the homing inhibition will persist even if the releases are stopped (while the dilution effect is manifested only during the releases). The dynamics can be very complex and counter-intuitive and dedicated modelling studies could describe in detail different scenarios. Please read also our common response #3.

To summarize, in the introduction the authors note that anti-crispr can be used to prevent spread in some contexts (left largely unspecified). The current manuscript shows that an anti-crispr can in fact be used to eliminate a HEG from the population, a significant achievement. What is missing, I think, is more context relating these results to earlier work, and contexts in which the strategies demonstrated in this manuscript (or other strategies) can be implemented. In short, I am just suggesting a bit more discussion and context creation to motivate the experiments carried out and results observed.

Thanks for this very valuable comment. In accordance, we thoroughly revised the manuscript as outlined at the beginning of the review response letter. In the revised version of the manuscript, we specified our motivation (lines 117-124) and outlined what should be the next steps of research to approach a potential field use of anti-drive strains (e.g. lines 669-677; 687-689). Please read also our more detailed common responses #1, #2 and #4.

One general comment on the modeling discussion. In the abstract the authors state that " A stochastic model predicted the experimentally-observed genotype dynamics in overlapping generations in medium- and large-sized cages and further demonstrated the effectiveness of anti-drive in different release and fitness scenarios. " However, in the modelling section and discussion the authors talk at length about the ways in which the model failed to capture behaviour when based on measurements of individual parameters. Would it make sense to change the abstract to something that indicates both strengths and limitations of modelling in environments with many complex unknowns? One take away I had from their work is that their results showed how much modelling needed to be informed by the results of experiments.

Bruce Hay

The abstract was modified as per reviewer's suggestion.

Reviewer #3 (Remarks to the Author):

D'amato et al. have developed a gene drive inhibitor system based on the AcrIIA4 strain, which is an anti-CRISPR molecule approach. Although their system is building of a previous AcrIIA4 system, their strain improved life-history fitness costs compared to previous constructs. They also develop a stochastic agent-based model to evaluate release parameters and fitness cost characteristics that determine frequency of anti-drive, gene drive and wild type alleles in large and medium cages. Overall, this is a well written article and it is the first demonstration of anti-drive impacts in large cage trials.

Thank you for this overall positive evaluation!

I have two specific comments that I would like addressed. One, I would like the authors to explain why there is a mismatch in model results compared to data from the cages - for example, with the time to establishment of populations varying between model and actual cage trials. More specifically, I would like the authors to explain what could be missing in their model to result in this mismatch because it is very likely models will be used to determine releases in field settings and a mismatch at the cage trial level may only lead to more uncertainty in larger simulations.

Thanks for this very valuable comment. We highlighted in the discussion now the limitation of the modelling to completely capture some of the biological complexity of age-structured populations, especially relative to the population stochasticity and the uncertainties to fully evaluate some fitness parameters (lines 592-595, 625-628). Please read our more detailed common response #3.

Second, could the authors comment on what happens when there are 'islands' of gene drive mosquitoes that can't be reached with anti-drive. Or are the authors proposing an optimal time of release (e.g. in the growth phase of gene drive spread) is all that's necessary to permanently inhibit gene drives in a population?

The present study focused on exploring the ability of an anti-crispr strain to control the spread in ecologically challenging (lab) settings. Population dynamics in field settings will largely depend on the ecology, geography, type and scale of the release and the interactions between interconnected or isolated populations. Regarding the specific examples, if gene drive mosquitoes reach 'isolated' populations where anti-drive can't reach, it is likely to induce local population collapse. Additional

more complex spatial models could show the dynamics over time in field-like settings. Please read also our more detailed common response #4.

(Minor comment) Scale-invariant probabilistic loss function:

- Typo at the end of sentence, "...structured overlapping populations that involve more complex behaviour and ecologically conditions."
- Uncommon citation style for references in text. For example "as described in 5 assessing egg deposition," Line 189 on page 4.

The typo is corrected and the format of the reference is now edited.

Reviewers' Comments:

Reviewer #2:

Remarks to the Author:

The authors have addressed my questions.

Bruce Hay